# TOKEN-LEVEL CORRELATION-GUIDED COMPRESSION FOR EFFICIENT MULTIMODAL DOCUMENT UNDERSTANDING

## ABSTRACT

Cropping high-resolution document images into multiple sub-images is the most widely used approach for current Multimodal Large Language Models (MLLMs) to do document understanding. Most of current document understanding methods preserve all tokens within sub-images and treat them equally. This neglects their different informativeness and leads to a significant increase in the number of image tokens. To perform a more adaptive and efficient document understanding, we propose Token-level Correlation-guided Compression, a parameter-free and plug-and-play methodology to optimize token processing. Firstly, we propose an innovative approach for assessing the pattern repetitiveness based on the correlation between each patch tokens. This method identifies redundant tokens, allowing for the determination of the sub-image's information density. Secondly, we present a token-level sampling method that efficiently captures the most informative tokens by delving into the correlation between the `[CLS]` token and patch tokens. By integrating these strategies, we develop a plug-and-play Token-level Correlation-guided Compressor module that can be seamlessly incorporated into MLLMs utilizing cropping techniques. This module not only enhances the processing speed during training and inference but also maintains comparable performance. We conduct experiments with the representative document understanding model mPLUG-DocOwl1.5 and the effectiveness is demonstrated through extensive comparisons with other compression methods.

## 1 INTRODUCTION

Document understanding is a vital and complex task that combines computer vision with natural language processing. The challenge arises from dealing with high-resolution document images with diverse aspect ratios, and parsing sparse or dense text with varied formats such as graphic or table. Recently, the rapid development of Multimodal Large Language Models (MLLMs) has demonstrated significant capabilities in image comprehension and instruction following (Bai et al., 2023; Dai et al., 2024; Liu et al., 2024a;b; Ye et al., 2023c; 2024; Zhang et al., 2024a). Some work further enhanced these models by incorporating high-resolution image processing and document parsing capabilities, leading to the development of sophisticated document understanding models (Feng et al., 2023; Hu et al., 2024; Liu et al., 2024c; Ye et al., 2023a;b).

Despite their impressive achievements, current MLLMs still struggle with efficient document understanding. As shown in Figure 1 (a), these models crop the original high-resolution image into multiple non-overlapping low-resolution sub-images (Dong et al., 2024; Hu et al., 2024; Liu et al., 2024c; Xu et al., 2024; Ye et al., 2023b). A large number of visual tokens are encoded by a vision encoder from all sub-images, and then collectively fed into a Large Language Model (LLM). This paradigm makes it hard for MLLMs to scale up to higher resolution documents as a dramatically growing number of visual tokens have to be dealt with. This significantly hinders the scalability and deteriorates the efficiency of current document understanding MLLMs.

To efficiently process high-resolution images, it is commonly believed that tokens within sub-images have different degrees of informativeness (Kong et al., 2022; Liu et al., 2024c; Zhang et al., 2024b), allowing for the compression of sub-images. Therefore, instead of simply feeding all tokens into

Figure 1: Comparison between (a) existing pipeline for cropping-based high-resolution processing methods and (b) proposed method. We can adaptively retain informative tokens, making models more efficient.

MLLMs, we could further delve into each sub-image and adaptively select the most informative tokens, as shown in Figure 1 (b). This would significantly decrease the number of tokens and contribute to a more efficient document understanding model. Based on the above idea, two challenges arise: **1)** how to adaptively determine the compression ratio for each sub-image, and **2)** how to design a adaptive compression strategy for sampling informative tokens.

To address these challenges, it is essential to measure the informativeness of each token. In this paper, we try to leverage the correlation between tokens to reflect the relative degree of informativeness. Specifically, we propose a Token-level Correlation-guided Compression method and explore the token-level correlations from two aspects, patch-patch and `CLS`-patch. **1) Using patch-patch correlation to determine the compression ratio**. We observe that some tokens within a sub-image exhibit repeated patterns and can be regarded as relatively less informative. To identify such less informative tokens, we investigate the patch-patch correlation to quantify the degree of the pattern repetitiveness regarding each token, and define the information density of a sub-image by the proportion of highly repetitive tokens. This information density could be subsequently utilized as the cues to determine compression ratio for each sub-image. **2) Exploiting `CLS`-patch correlation to sample tokens**. The `[CLS]` token can aggregate and summarize descriptive global information of an image when it exhibits a higher correlation with the informative patch tokens. Consequently, we might detect and sample the most informative patch tokens based on the correlation between `[CLS]` and patch tokens. Based on this idea, to effectively sample the most informative tokens, we analyze the `CLS`-patch correlation and form a probability distribution to guide the sampling process.

Through the guidance of token-level correlation, we construct a plug-and-play high-resolution Token-level Correlation-guided Compressor module. It can be applied as plug-and-play to high-resolution MLLMs that use cropping methods, improving training and inference speed with slight or even no performance loss. We conduct experiments with mPLUG-DocOwl1.5 (Hu et al., 2024), which is a representative cropping-based model in document understanding. Experimental results indicate that the proposed method can maintain comparable performance with DocOwl1.5 while achieving a maximum compression ratio of 11.5%. Further extensive ablation experiments also verify the effectiveness of the method. Our contributions are summarized as follows:

• We conduct an in-depth study of patch-patch correlation to quantify the pattern repetitiveness of tokens and introduce the concept of information density of sub-image. Furthermore, we reveal the distribution of informative tokens with the guidance of `CLS`-patch correlation and propose a strategy to effectively sample informative tokens based on the distribution.

• We propose the Token-level Correlation-guided Adaptive Compression method. It aims to adaptively compress visual token according to the information density. This adaptation allows for more efficient understanding of document with diverse types by fitting them into different lengths.

• Comprehensive experiments demonstrate that the proposed approach can significantly compress image tokens while maintaining comparable performance. The proposed method achieve an average compression ratio of 66% across different datasets, with a maximum ratio of up to 11.5%, significantly enhancing the model's efficiency.

## 2 RELATED WORK

### 2.1 DOCUMENT UNDERSTANDING

To enable models to understand document images, a major challenge is the ability to process high-resolution images. There are currently two main methods for processing: one is to use heuristic crop-

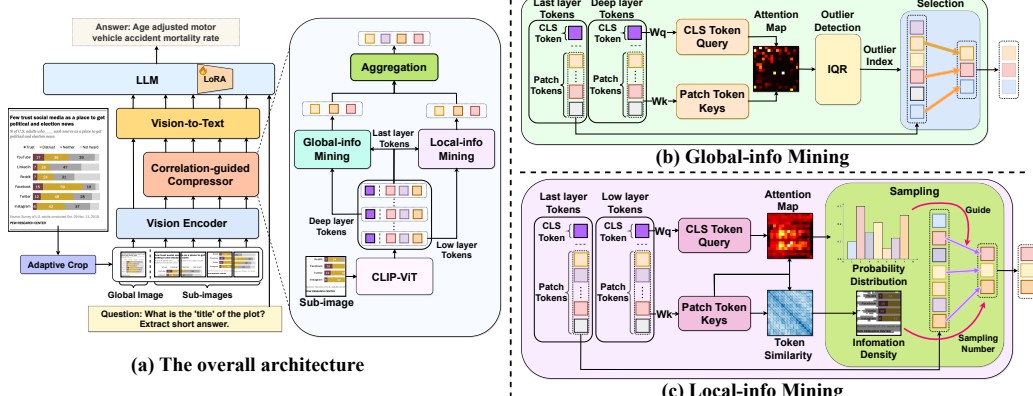

Figure 2: The illustration of the proposed method. (a) The overall architecture. The Token-level Correlation-guided Compressor is inserted between vision encoder and vision-to-text, which comprises two branches, (b) global information mining branch and (c) local information mining branch.

ping (Liu et al., 2024d; Luan et al., 2024; Wu & Xie, 2024; Zhang et al., 2023), and another is to crop high-resolution image to a size that can be properly recognized by vision encoder. Pix2Struct (Lee et al., 2023) first proposes a variable-resolution input representation for document understanding. While exhibiting promising ability in high-resolution perception, its language comprehension ability is much limited by the usage of a lightweight language decoder. And its vision encoder needs to be trained from scratch, unable to utilize the existing pre-trained model. To draw these issues, UReader (Ye et al., 2023b) further proposes a shape-adaptive cropping module to crop raw images into multiple non-overlapping sub-images with low-resolution that fit the size of pre-trained vision encoder, and conducts initial exploration on fine-tuning document understanding tasks based on MLLMs. Due to the strong high-resolution perception ability and language understanding ability demonstrated by UReader, the cropping-based high-resolution processing method has been widely adopted by subsequent works. For example, Monkey (Li et al., 2024) employs a sliding window technique to crop image and TextMonkey (Liu et al., 2024c) further introduces a shifted window attention mechanism to enable the interaction between different sub-images. mPLUG-Docowl1.5 (Hu et al., 2024) utilizes the same shape-adaptive cropping module as UReader and strengthens the document understanding ability by unified structure learning, achieving state-of-the-art performance. Despite their robust capabilities for document understanding, these models remain significantly inefficient. We propose token-level correlation-guided compression to enhance the efficiency of document understanding in MLLMs.

## 2.2 TOKEN COMPRESSION

Introducing high-resolution images into MLLM will significantly increase the number of visual tokens, necessitating methods to reduce the length of visual token sequences for efficient training and inference. Most existing works choose to use vision-to-text modules with compression capabilities, including using learnable queries along with cross attention mechanism (Bai et al., 2023; Dai et al., 2024; Liu et al., 2024c; Ye et al., 2023b;c; 2024), convolutional layer with strides (Hu et al., 2024; Lu et al., 2024), or simply concatenate the adjacent tokens into a single new token via the channel dimension (Dong et al., 2024). Although these methods demonstrate promising token compression ability, when it comes to cropping-based high-resolution MLLMs, they are still not efficient due to the different informativeness of different tokens. Previous studies have explored the token compression for efficient transformer processing within the fields of natural language processing (Goyal et al., 2020; Kim & Cho, 2021; Kim et al., 2022; Lassance et al., 2021) and computer vision (Meng et al., 2022; Rao et al., 2021; Song et al., 2022; Yin et al., 2022; Yu & Wu, 2023) independently. However, to our knowledge, investigations into token compression in the multimodal domain remain relatively limited. Concurrently, Shang et al. (2024) propose a PruMerge algorithm to adaptively select unpruned visual tokens based on their similarity to class tokens and spatial tokens. TextMonkey (Liu et al., 2024c) has proposed a Token Filter module that retains the most unique visual tokens for compression. However, they do not thoroughly consider token-level correlations from different perspectives, which limits their ability to adaptively compress visual tokens for diverse data types and their capacity to maintain performance.

Figure 3: Visualization of token similarity. We select tokens corresponding to visually repetitive patches and visualize the similarity between the selected tokens and others. It can be observed that visually repetitive patches exhibit a high degree of similarity between their corresponding tokens.

## 3 PROPOSED METHOD

The overall architecture of the proposed method is presented in Figure 2. Following previous works (Hu et al., 2024; Liu et al., 2024c; Ye et al., 2023b), the model begins by cropping high-resolution input image into multiple non-overlapping sub-images that fit the pre-training size of vision encoder. All sub-images along with the resized global image are fed into the vision encoder to get visual tokens. In previous methods, the visual token sequences are aligned with text inputs via a vision-to-text module. Then all of them would be concatenated with text tokens and collectively fed into LLM for processing, which is extremely inefficient in the face of high-resolution document images. In this work, we introduce Token-level Correlation-guided Compressor module to adaptively compress visual tokens. The Token-level Correlation-guided Compressor first uses an information density calculation module to adaptively decide the compression ratio of each sub-image, which will be discussed in section 3.1. After that, this module utilizes a correlation-guided token sampling method to sample the most informative tokens, which will be discussed in section 3.2. The workflow of Token-level Correlation-guided Compressor module will be detailed in section 3.3.

### 3.1 PATCH-PATCH CORRELATION-GUIDED INFORMATION DENSITY CALCULATION

Document images often contain large areas of whitespace and color blocks, which visually present repetitive patterns and can be considered relatively less informative and redundant for understanding the image. To determine an appropriate compression ratio for a sub-image, it is necessary to identify the redundant areas and reflect the proportion of unique regions within the image.

Since patch tokens typically contain local information within the image, we believe that the patch tokens corresponding to visually repetitive patches are highly correlated. This inspires us to explore patch-patch correlation for identifying redundant tokens. Specifically, we utilize the keys vectors from attention layers in CLIP to represent each token, denoted as $K \in R^{N \times D_k}$, where $N$ represents the number of visual tokens in a image and $D_k$ represents the dimension. We then calculate the cosine similarity between pairs of tokens (Bolya et al., 2022) by the following formula:

$$S_{ij} = \frac{K_i K_j}{\|K_i\|\|K_j\|}, i, j \in [1, N] \tag{1}$$

As shown in Figure 3, it can be observed that the tokens corresponding to visually repetitive patches in the image have many highly similar counterparts, which verifies our hypothesis. This finding enables the differentiation between redundant patch tokens from others.

Based on this finding, we design a method to adaptively calculate the proportion of non-redundant tokens in a sub-image, termed **information density**. Specifically, we calculate the cosine similarity between pairwise tokens $S \in R^{N \times N}$ using Equation 1. For a given token, if the number of tokens with a similarity greater than the threshold $\alpha$ exceeds an upper limit $k$, it will be regarded as redundant. We calculate the proportion of redundant tokens in the sub-image to the total number of tokens as the information redundancy, noted as $r$, and $d = 1 - r$ represents the information density. Finally, we treat the information density as the compression ratio of each sub-image. The patch-patch correlation-guided information density calculation is detailed in Algorithm 1.

### 3.2 CLS-PATCH CORRELATION-GUIDED INFORMATIVE TOKEN SAMPLING

In CLIP-ViT (Radford et al., 2021), a special [CLS] token is introduced to aggregate information from each patch token, creating a representation of the entire image. Consequently, there should be

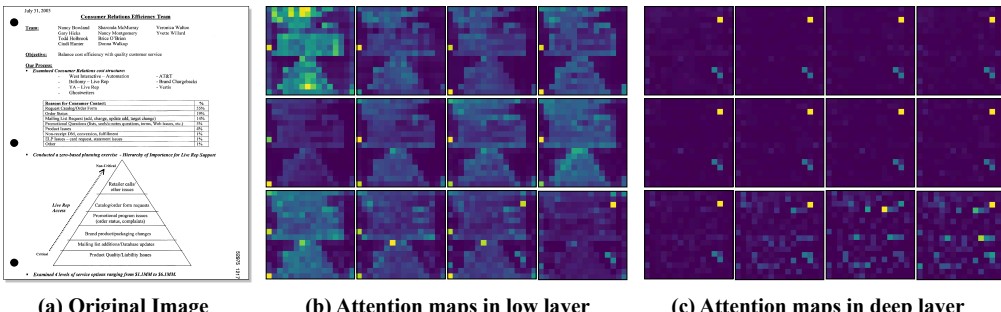

|(a) Original Image|(b) Attention maps in low layer|(c) Attention maps in deep layer|

Figure 4: Visualization of the attention maps between the `[CLS]` token and patch tokens across different layers of CLIP-ViT-L. (a) Original input image. (b) The attention maps from layers 1 to 12. (c) The attention maps from layers 13 to 24.

a higher correlation between the `[CLS]` token and the informative tokens. This aggregation process is achieved through the self-attention mechanism, which can be formulated as:

$$\text{Self-Attention}(Q, K, V) = \text{Softmax}(\frac{QK^T}{\sqrt{D_k}})V \tag{2}$$

where value $V \in R^{N \times D_v}$, query and key $Q, K \in R^{N \times D_k}$ are three individual vectors projected from the visual tokens $Y \in R^{N \times D}$. The attention map $A = \text{Softmax}(\frac{QK^T}{\sqrt{D_k}}) \in R^{N \times N}$ serves as a weight matrix for aggregating information from other tokens, thus inherently indicating the extent of correlation between tokens. To measure the informativeness of each token, we interpret the attention scores between `[CLS]` token and patch tokens as a quantification of the relative informativeness of patch tokens, with higher scores indicating that a patch token is relatively more informative.

To unveil the underlying patterns of the `CLS`-patch correlation presented by the self-attention mechanism, we conduct an in-depth study of the attention maps across different layers of CLIP-ViT. As shown in Figure 4, the low layer attention maps highlight the visually informative regions in the image, aligning with the consensus that patch tokens hold local information. Nevertheless, the attentions in deep layers exhibit several irregular outliers. This phenomenon can be attributed to the model's tendency to gradually integrate image information at multiple locations, facilitating the aggregation of global information into the `[CLS]` token. As suggested by Darcet et al. (2023), the outlier can be considered to hold global information while containing minimal local information. These distribution patterns of `CLS`-patch correlation in both deep and low layers inspire a potential approach to identify tokens that contain both global and local information. More visualization results can be found in **Appendix A.3.1**.

---

**Algorithm 1:** Information Density Calculation

---

**Data:** Key matrix of attention from CLIP-ViT, $K \in R^{N \times D}$. Threshold of similarity, $\alpha$. Upper limit of similarity tokens $k$. The number of tokens $N$. The dimension of embeddings $D$.

**Result:** The information density $d$ of the sub-image.

1. Normalize each key vector of attention: $K_i \leftarrow \frac{K_i}{\|K_i\|}, i = 1, 2, \cdots, N$
2. Calculate the similarity between pairwise tokens: $S = KK^T \in R^{N \times N}$
3. Initialize the number of redundant tokens with: $N_R \leftarrow 0$.
4. **for** $i = 1$ *to* $N$ **do**
5.    $n \leftarrow 0$
6.    **for** $j = 1$ *to* $N$ **do**
7.       **if** $S_{i,j} > \alpha$ **then**
8.          $n \leftarrow n + 1$
9.       **end**
10.    **end**
11.    **if** $n > k$ **then**
12.       $N_R \leftarrow N_R + 1$
13.    **end**
14. **end**
15. Calculate information redundancy: $r = \frac{N_R}{N}$, and information density: $d = 1 - r$

---

To effectively sample informative tokens from sub-images, we design methods that respectively sample tokens with the guidance of `CLS`-patch correlation from both deep and low layers. Leveraging the `CLS`-patch correlation from deep layers, we adopt the Interquartile Range (IQR) method to distinguish outliers (Boukerche et al., 2020) for preserving informative tokens with global information. Specifically, IQR calculates the first quartile (Q1) and the third quartile (Q3) in a set of numbers. It considers values that are 1.5 times the IQR above Q3 as upper outliers and values that are 1.5 times the IQR below Q1 as lower outliers. As a typical outlier recognition method, IQR can adapt well to different data distributions and determine an adaptive number of outliers. We preserve tokens identified as upper outliers, thereby retaining informative tokens with global information.

To further preserve informative tokens with local information, we quantify the relative informativeness of different tokens through the attention map from low layer. Since the attention map only displays relative relationships, directly setting a threshold is not feasible. Instead, we treat the attention score as the probability of sampling each token and perform sampling without replacement. This method ensures that more informative tokens are more likely to be selected.

### 3.3 TOKEN-LEVEL CORRELATION-GUIDED COMPRESSION

We propose a plug-and-play Token-level Correlation-guided Compressor module by integrating the information density calculation and informative tokens sampling. As shown in Figure 2, the module consists of two parallel branches, the global token mining branch and the local token mining branch. The global token mining branch focuses on selecting and preserving tokens that contain global information. It utilizes the `CLS`-patch correlation in the deep layers of CLIP-ViT to adaptively retain an appropriate number of tokens through IQR. Meanwhile, the local token mining branch aims to sample tokens that hold local information. To determine the sampling ratio adaptively, we calculate the information density of the sub-image with the guidance of patch-patch correlation and use it as the sampling ratio. To effectively sample the most informative tokens, the `CLS`-patch correlation in the low layers of CLIP-ViT is leveraged to form the sampling distribution. Consequently, tokens are sampled without replacement based on the token-level correlation-guided sampling ratio and distribution. All sampled tokens are then concatenated as the preserved tokens.

---

**Algorithm 2:** Token-level Correlation-guided Compression

**Data:** The output visual tokens from the last layer, selected deep layer and low layer of CLIP-ViT, $Y, Y_d, Y_l \in R^{N \times D}$. $N$ is the number of visual tokens and $D$ is dimension.

**Result:** Compressed visual tokens $Y' \in R^{n \times D}$, where $n < N$.

1 **(Preliminary)** Calucate attention key, query matrix and attention map of selected deep layer $Q_d, K_d, A_d$, and selected low layer $Q_l, K_l, A_l$

2 **(Global info mining)** Adaptively select $s$ token indices $I = \{i_1, \cdots, i_s\}$ using the IQR algorithm based on $A_d$.

3 **(Local info mining)** Calucate information density $d$ using $K_l$.

4 **(Local info mining)** Use $A_l$ as the probability distribution and perform sampling without replacement, obtaining $d' = \lfloor d \times N \rfloor$ tokens indices $J = \{j_1, \cdots, j_{d'}\}$.

5 **(Aggregation)** Merge $I$ and $J$, remove duplicates and sort, resulting $L = \{l_1, \cdots, l_n\}$.

6 **(Aggregation) for** $l$ *in* $L$ **do**

7      Calculate the distance between selected token $y_l$ and other tokens $y_{\{1,\cdots N\}/l}$;

8      Use $k$-nearest neighbor algorithm to find $k$ similar tokens, with indices $P = \{p_1, \cdots, p_k\}$;

9      Update $y_l$ by weighted sum: $y'_l = \sum_{p \in P} A_{d,p} \cdot y_p$.

10 **end**

---

To prevent the unintended discarding of important information, we adopted a token aggregation method as an alternative to directly discarding unsampled tokens (Bolya et al., 2022; Marin et al., 2021). Specifically, this method utilizes keys of attention to calculate the similarity between tokens as a distance metric. Each sampled token is grouped with other tokens using k-nearest neighbors. Then the representations of each group are updated through a weighted sum, with attention scores serving as the weights. A similar approach has also been employed by Shang et al. (2024). Finally, the aggregated tokens are fed into the vision-to-text module and LLM as a compressed input. Specifically, to keep the overall information of the image, we do not compress the global image. The complete procedure of the Token-level Correlation-guided Compression is detailed in Algorithm 2.

Table 1: Comparison with existing document understanding models. The scores marked with an underline represent the state-of-the-art (SOTA) performance. Although the proposed method exhibits a slight degradation compared to the SOTA, it achieves an average compression rate of 66% across various datasets, enhancing efficiency significantly. * The methods of PruMerge, PruMerge+ and Token Filter are both tested with DocOwl1.5 in plug-and-play mode. † We replace the global and local information mining modules in our proposed method with the Token Filter module from TextMonkey and the compression ratio is set to 50%.

| Model | Doc VQA | Info VQA | Deep Form | KLC | WTQ | Tab Fact | Chart QA | Text VQA | Text Caps | Visual MRC |
|---|---|---|---|---|---|---|---|---|---|---|
| | **Without compression** | | | | | | | | | |
| DocPeida (Feng et al., 2023) | 47.1 | 15.2 | - | - | - | - | 46.9 | 60.2 | - | - |
| Monkey (Li et al., 2024) | 66.5 | 36.1 | 40.6 | 32.8 | 25.3 | - | - | 67.6 | 93.2 | - |
| DocOwl (Hu et al., 2024) | 62.2 | 38.2 | 42.6 | 30.3 | 26.9 | 60.2 | 57.4 | 52.6 | 111.9 | 188.8 |
| UReader (Ye et al., 2023b) | 65.4 | 42.2 | 49.5 | 32.8 | 29.4 | 67.6 | 59.3 | 57.6 | 118.4 | 221.7 |
| CogAgent (Hong et al., 2024) | 81.6 | 44.5 | - | - | - | - | 68.4 | 76.1 | - | - |
| TextMonkey (Liu et al., 2024c) | 71.5 | - | 61.6 | 37.8 | 30.6 | - | 65.5 | 68.0 | - | - |
| DocOwl1.5 (Hu et al., 2024) | 81.6 | 50.4 | 68.8 | 37.9 | 39.8 | 80.4 | 70.5 | 68.8 | 132.0 | 239.5 |
| | **Compression without considering token-level correlation** | | | | | | | | | |
| *Prumerge (Shang et al., 2024) | 53.6 | 29.6 | 9.3 | 28.3 | 23.7 | 71.4 | 55.8 | 60.0 | 120.7 | 125.9 |
| *Prumerge+ (Shang et al., 2024) | 55.0 | 33.2 | 26.7 | 30.8 | 24.6 | 70.9 | 58.3 | 62.3 | 124.8 | 185.0 |
| *†Token Filter (Liu et al., 2024c)[1] | 69.9 | 41.7 | 52.9 | 35.8 | 30.5 | 72.5 | 65.1 | - | - | 209.9 |
| | **Adaptive Compression guided by token-level correlation** | | | | | | | | | |
| Ours (plug-and-play) | 72.6 | 49.6 | 63.2 | 34.6 | 35.2 | 75.2 | 64.0 | **68.0** | **132.1** | **254.3** |
| Ours (finetuning) | **78.3** | **50.2** | **65.7** | **35.9** | **38.6** | **79.3** | **68.9** | 66.6 | 125.9 | 243.7 |

Through the token-level correlation-guided compression, all sub-images are compressed into different lengths adaptively. The method maximally retains information while concurrently minimizing the number of visual tokens for all sub-images, thus significantly enhancing model efficiency. We would like to emphasize the distinction between the proposed method and the Token Filter introduced by TextMonkey Liu et al. (2024c). While the Token Filter relies on similarity to identify tokens for pruning, requiring manual specification of how many tokens to remove, the proposed method uses similarity to assess information density, allowing for an adaptive determination of the compression ratio. Additionally, the Token Filter assumes that tokens with similar counterparts carry less information and prunes them directly. In contrast, the proposed approach focuses on mining informative tokens based on the CLS-patch correlation. Experimental results in Section 4.1 highlight the superiority of the proposed method.

## 4 EXPERIMENTS

We conduct experiments with mPLUG-DocOwl1.5 (Hu et al., 2024). For more implementation details, please refer to **Appendix A.1**.

### 4.1 EXPERIMENTAL RESULTS

We evaluate the proposed method on 10 datasets with diverse types, including text-rich datasets like DocVQA (Mathew et al., 2021), InfoVQA (Mathew et al., 2022), DeepForm (Svetlichnaya, 2020), KLC (Stanisław et al., 2021), table datasets like WTQ (Pasupat & Liang, 2015), TabFact (Chen et al., 2019), chart datasets like chartQA (Masry et al., 2022), natural datasets like TextVQA (Singh et al., 2019), TextCaps (Sidorov et al., 2020), and webpage screenshots dataset VisualMRC (Tanaka et al., 2021). The results are compared with previous OCR-free methods (Feng et al., 2023; Hong et al., 2024; Li et al., 2024; Liu et al., 2024c; Ye et al., 2023a;b) and other token compression methods (Shang et al., 2024; Liu et al., 2024c).

---

[1]Due to the official evaluation portals for TextVQA and TextCaps Challenges being unavailable at the time of writing, we are unable to provide the results for Token Filter.

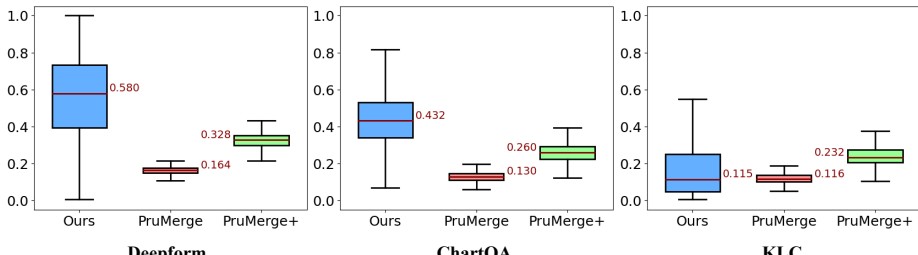

Figure 5: Boxplot visualization of the compression ratio achieved by different compression methods across various datasets. The median numbers is presented adjacent to the boxes.

Table 2: We conducted an experiment to estimate the inference speed improvement of DocOwl1.5 after incorporating our method. 100 samples are randomly selected from each dataset and the total inference time is recorded. On average, the speed has improved by 13.0%.

| Dataset | DocVQA | InfoVQA | Deepform | KLC | ChartQA | VisualMRC |
|---|---|---|---|---|---|---|
| Docowl-1.5 | 77.04s | 57.56s | 95.24s | 94.27s | 58.20s | 124.53s |
| Ours | 66.88s | 52.21s | 87.82s | 79.03s | 44.78s | 113.79s |
| Speed Improvement ↑ | **13.2%** | **9.3%** | **7.8%** | **16.2%** | **23.1%** | **8.6%** |

As shown in Table 1, our method performs better than many previous OCR-free methods. Compared to the baseline model DocOwl1.5, the proposed method achieves comparable performance in plug-and-play mode, with an average compression ratio of 66%. In contrast, PruMerge, PruMerge+ algorithm (Shang et al., 2024) and Token Filter module from TextMonkey (Liu et al., 2024c) result in significant performance degradation.

We further investigate the token compression ratio of different adaptive compression methods on various datasets. Each cropped sub-image is treated as an individual sample, and we calculate the compression ratio for all sub-images. Here, the compression ratio is defined as the number of tokens after compression divided by the original number of tokens. A lower compression ratio indicates a more effective compression. As shown in Figure 5 and Figure 6, on different datasets, the compression ratios resulted from PruMerge and PruMerge+ (Shang et al., 2024) algorithms remain within a relatively fixed interval, while the proposed method exhibits significantly different compression ratios on different datasets. This result demonstrates that the proposed method can adaptively recognize the information distribution patterns of different datasets and identify the most appropriate compression ratio. More results on different datasets can be found at **Appendix A.2.2**. As a summary, the proposed method achieves an average compression ratio of 66% across different datasets, with a maximum ratio of up to 11.5%, significantly enhancing the model's efficiency.

To validate the acceleration effects of the proposed method in practical applications, we conduct experiments to assess the speed improvement. As shown in Table 2, after integrating the proposed method, Docowl 1.5 Hu et al. (2024) achieves an average inference speed increase of 13% across different datasets. The acceleration effects varied among the datasets, further demonstrating the adaptive compression capability of the proposed method for different data distributions. Note that by adjusting the hyperparameters $\alpha$ and $k$, the acceleration effects can be further enhanced, as detailed in the ablation results in **Appendix A.2.4**.

## 4.2 ABLATION STUDY

We perform a comprehensive ablation study to validate the effectiveness of `CLS`-patch correlation-guided token sampling, information density-based sampling ratio and global/local information mining. These experiments are conducted in plug-and-play mode for comparison.

**Effectiveness of `CLS`-patch correlation-guided token sampling**. For the sampling strategy in local information mining, we compare the `CLS`-patch correlation-guided sampling with uniform and random sampling under the same sampling ratio. As shown in Table 3, the token correlation-guided sampling significantly outperforms both uniform and random sampling.

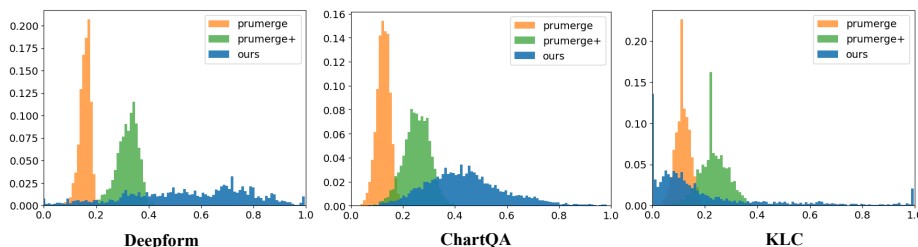

Figure 6: Histogram visualization of the compression ratio achieved by different compression methods across various datasets.

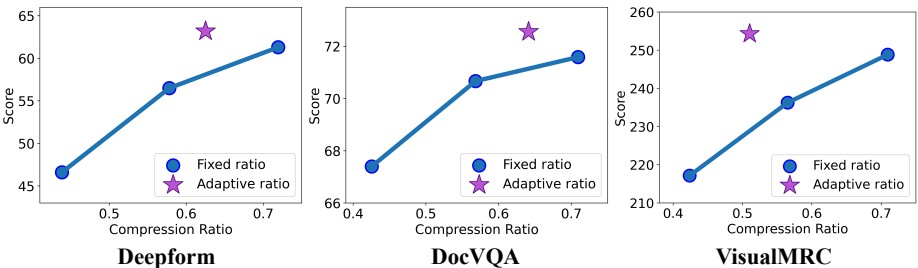

Figure 7: Comparisons between our adaptive compression ratio and fixed compression ratios. We set the sampling ratio at multiple fixed values and tested these on various datasets. Despite the higher average compression ratios resulted by fixed ratios, the evaluation scores are unable to surpass those of our adaptive compression method.

Table 3: Comparison between different methods in local information mining branch.

| Sampling Method | Deep Form | Doc VQA | Visual MRC |
|---|---|---|---|
| Random | 55.8 | 69.1 | 245.4 |
| Uniform | 56.6 | 70.0 | 248.8 |
| Ours | **63.2** | **72.6** | **254.3** |

Table 4: Effectiveness of the global and local information mining branch.

| Global Info Mining | Local Info Mining | Deep Form | Doc VQA | Visual MRC |
|---|---|---|---|---|
| $\times$ | $\checkmark$ | 61.0 | 71.7 | 249.7 |
| $\checkmark$ | $\times$ | 9.3 | 53.6 | 125.9 |
| $\checkmark$ | $\checkmark$ | **63.2** | **72.6** | **254.3** |

**Effectiveness of information density calculation**. We also conduct another set of experiments to verify the effectiveness of our adaptive sampling ratio. We set a group of fixed sampling ratios of $\frac{2}{3}$, $\frac{1}{2}$ and $\frac{1}{3}$ in local information mining for comparison. As shown in Figure 7, for a fixed sampling ratio setting, even though the fixed sampling ratio settings retain more tokens on average, its performance still cannot surpass our adaptive sampling ratio method.

**Effectiveness of global and local information mining.** In Table 4, we conduct ablation experiments on two modules: global information mining and local information mining. As shown in table 4, simply removing any part here will result in performance degradation.

In order to validate that attention maps from low layers of CLIP-ViT can effectively guide the sampling of informative tokens, we conduct experiments to examine the selection of attention maps from different layers in local information mining. The results can be found in **Appendix A.2.3**.

Furthermore, we conduct experiments to analyze the impact of adjusting the hyperparameters $\alpha$ and $k$ in the calculation of information density. The results are provided in **Appendix A.2.4**. In summary, by adjusting $\alpha$ and $k$, the proposed method can further enhance efficiency through more intense compression or better maintain performance by retaining more tokens, thereby achieving a desired balance between efficiency and performance.

## 4.3 VISUALIZATION

To intuitively verify the effectiveness of the proposed method, we conduct several visualization experiments. We first visualize the redundant tokens identified during information density calculation.

As shown in Figure 8, the unmasked areas indicate the locations of identified redundant tokens, which are all concentrated in the visually repetitive regions. These results validate the effectiveness of our information density calculation method. Simultaneously, the calculated information density accurately reflects the relative degree of informativeness of different sub-images, thereby demonstrating the effectiveness of our adaptive compression capability. More visualization results of the information density calculation on different datasets can be found at **Appendix A.3.2**.

We also visualize the tokens sampling results achieved by the `CLS`-patch correlation-guided sampling method. Several samples with different distribution patterns are selected to verify the effectiveness of our method. In Figure 9, it can be observed that tokens sampled by local information mining are mainly concentrated in the area of informative areas, which verifies the effectiveness of our method. More results can be found at **Appendix A.3.3**.

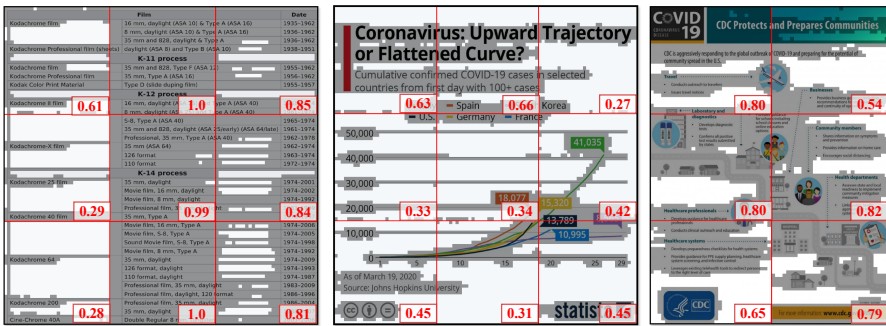

Figure 8: Visualization of redundant patch tokens. We visualize the redundant tokens identified during the information density calculation, as represented by the unmasked areas. The calculated information density is highlighted at the bottom right of each sub-image. It can be observed that the identified redundant tokens are concentrated in the parts that visually present repetitive patterns.

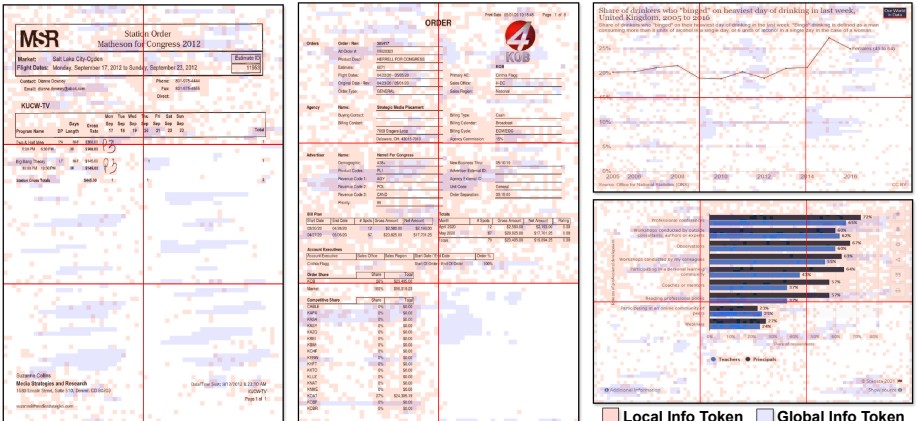

Figure 9: Visualization of selected token. The blue parts represent the tokens preserved by global information mining, while the orange parts represent the sampled tokens in local information mining.

## 5 CONCLUSION

In this paper, we present a token-level correlation-guided compression method to enhance document understanding efficiency in MLLMs. Experimental results show significant token sequence length reduction while maintaining performance comparability. The proposed method still has some limitations, including the necessity for fine-tuning the model to minimize the performance disparity with the base model. Additionally, a deterministic sampling method can be more robust to diverse applications. We hope to address these issues in future work.

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

# A APPENDIX

## A.1 IMPLEMENTATION DETAILS

We conduct experiments with the mPLUG-DocOwl1.5 model (Hu et al., 2024), which utilizes the CLIP-ViT/L-14 (Radford et al., 2021) as the vision encoder and 7B LLaMA-2 model (Touvron et al., 2023) with the Modality Adaptive Module (Ye et al., 2024) as the language decoder. We keep the adaptive cropping method in mPLUG-DocOwl1.5 with a fixed resolution of $448 \times 448$. The similarity threshold $\alpha$ and upper limit $k$ in information density calculation is set to $0.7$ and $50$, respectively. We choose the last layer and the eighth layer to guide global and local information mining. Note that our method is a parameter-free token compression method. Thus we mainly conduct our experiment in the plug-and-play manner. We also further finetune mPLUG-Docowl.5 with our method using LoRA (Hu et al., 2021) and DocDownstream dataset (Hu et al., 2024) for 1 epoch. The initial learning rate for fine-tuning is set to 1e-4, with a batch size of 256. The vision-to-text module is frozen, and only the LoRA parameters are trained. We conduct experiments on a server equipped with four Nvidia A800 GPUs, each with 80GB of VRAM.

## A.2 MORE QUANTITATIVE EXPERIMENTS

### A.2.1 STATISTICAL SIGNIFICANCE ANALYSIS

To mitigate the impact of randomness introduced by the sampling used in the proposed method, we conducted three repeated experiments across multiple datasets. As shown in Table 5, the random errors introduced by sampling are minimal. Even accounting for these errors, the method still demonstrates significant superiority over PruMerge and PruMerge+ (Shang et al., 2024).

Table 5: Replicate experiments for mitigating the impact of sampling.

| Experiment | DeepForm | KLC | WTQ | ChartQA |
|---|---|---|---|---|
| PruMerge | 9.33 | 28.25 | 23.67 | 55.84 |
| PruMerge+ | 26.72 | 30.84 | 24.55 | 58.28 |
| Ours | **62.80±0.29** | **34.54±0.02** | **35.32±0.17** | **64.38±0.39** |

### A.2.2 COMPARISON TO THE COMPRESSION RATIO

In this section, we present more results of the statistically adaptive compression ratio across additional datasets. As shown in Figure 11 and Figure 12, the PruMerge and PruMerge+ algorithms maintain relatively fixed compression ratios across various datasets. In contrast, our method adapts the compression rate based on the distribution pattern of the data, resulting in significant variations across different datasets.

### A.2.3 ABLATION OF LAYER SELECTION

To demonstrate that attention maps from low layers of CLIP-ViT can effectively guide the sampling of informative tokens, we conduct experiments to evaluate the use of attention maps from different layers in local information mining. Specifically, for low layers, we select layers 4 and 8, and for deep layers, we select the 16th and 24th layers. As shown in Table 6, the performance of selecting attention maps from the 4th and 8th layers outperforms that of the 16th and 24th layers, demonstrating that low layer attention maps more accurately measure the distribution of informative tokens containing local information. Additionally, the superior performance of the 8th layer compared to the 4th suggests that shallower layers are not always better. This may be due to the inability of tokens in the very shallow layers to form descriptive representations. Moreover, the performance drops significantly when selecting the 24th layer compared to the 20th layer, further indicating that deep layers tend to describe the distribution of tokens with global information.

Table 6: Comparisons of choosing different layers to guide local information mining

| Layer | DeepForm | DocVQA | VisualMRC |
|-------|----------|--------|-----------|
| 4 | 61.6 | 71.5 | 251.7 |
| 16 | 60.9 | 70.9 | 250.8 |
| 24 | 52.9 | 66.8 | 247.0 |
| 8 | **63.2** | **72.6** | **254.3** |

### A.2.4 ABLATION OF HYPERPARAMETERS

We provide the ablation results of hyperparameters $\alpha$ and $k$ in Figure 10. In the proposed method, the average compression ratio can be controlled by adjusting the $\alpha$ and $k$. Generally, preserving more tokens lead to greater accuracy. This means that by adjusting $\alpha$ and $k$, a desired balance between efficiency and accuracy can be achieved.

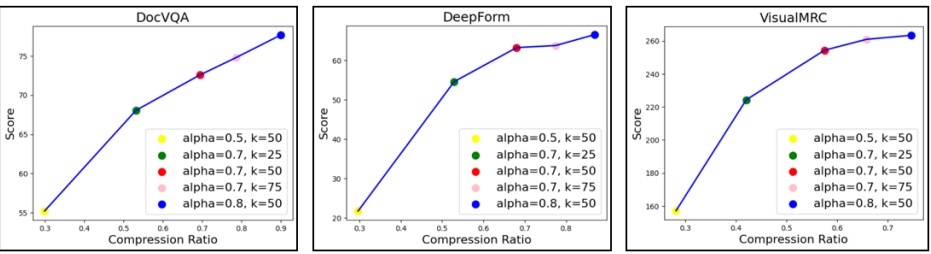

Figure 10: The ablation study of hyperparameters $\alpha$ and $k$. The red dots represent the settings we used in our paper.

## A.3 MORE VISUALIZATION RESULTS

### A.3.1 ATTENTION MAPS ACROSS DIFFERENT LAYERS OF CLIP-ViT

We show more visualization results of attention maps across different datasets, including various types of tables, text-rich, and charts. As depicted in Figure 13, the attention maps generated by CLIP-ViT exhibit the same distribution patterns in various types of data, which further corroborates our findings.

### A.3.2 REDUNDANT PATCH TOKEN VISUALIZATION

In this section, we will present more visualization results about redundant patch tokens. As shown in Figure 14, the information density calculated by our method is consistent with the visual perception.

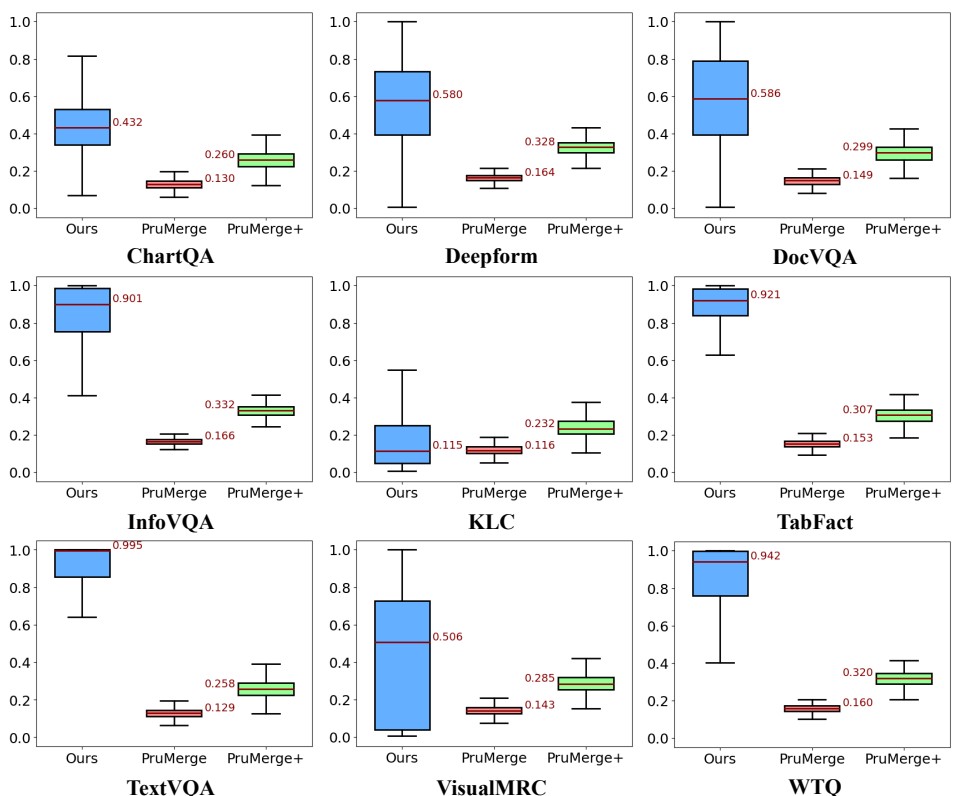

Figure 11: Boxplot visualization of the compression ratio achieved by different compression methods across additional datasets. The median numbers is presented adjacent to the boxes.

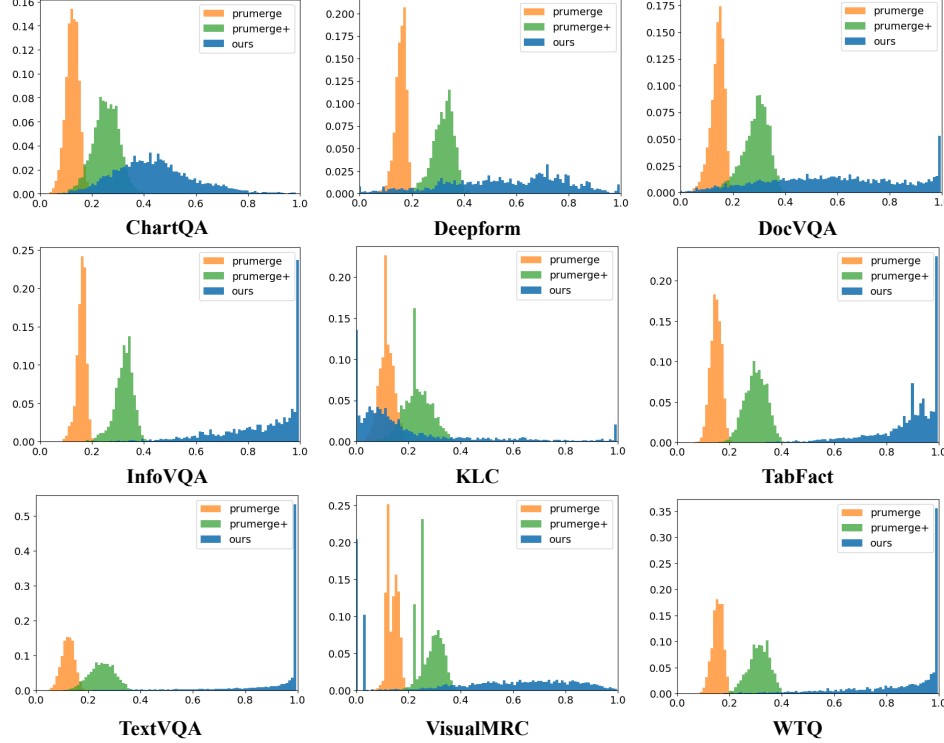

Figure 12: Histogram visualization of the compression ratio achieved by different compression methods across additional datasets.

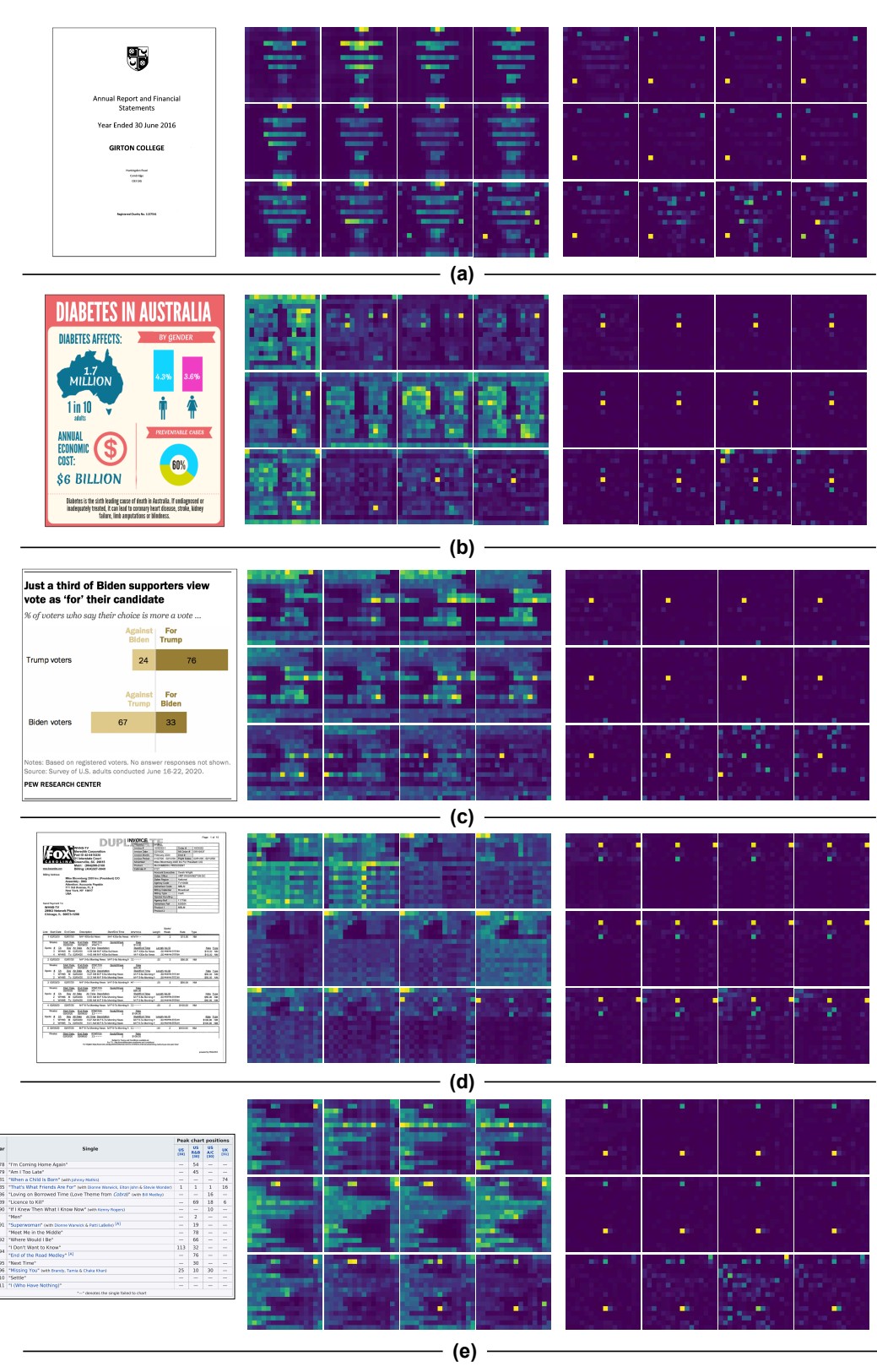

Figure 13: More visualization results of attention maps in CLIP-ViT.

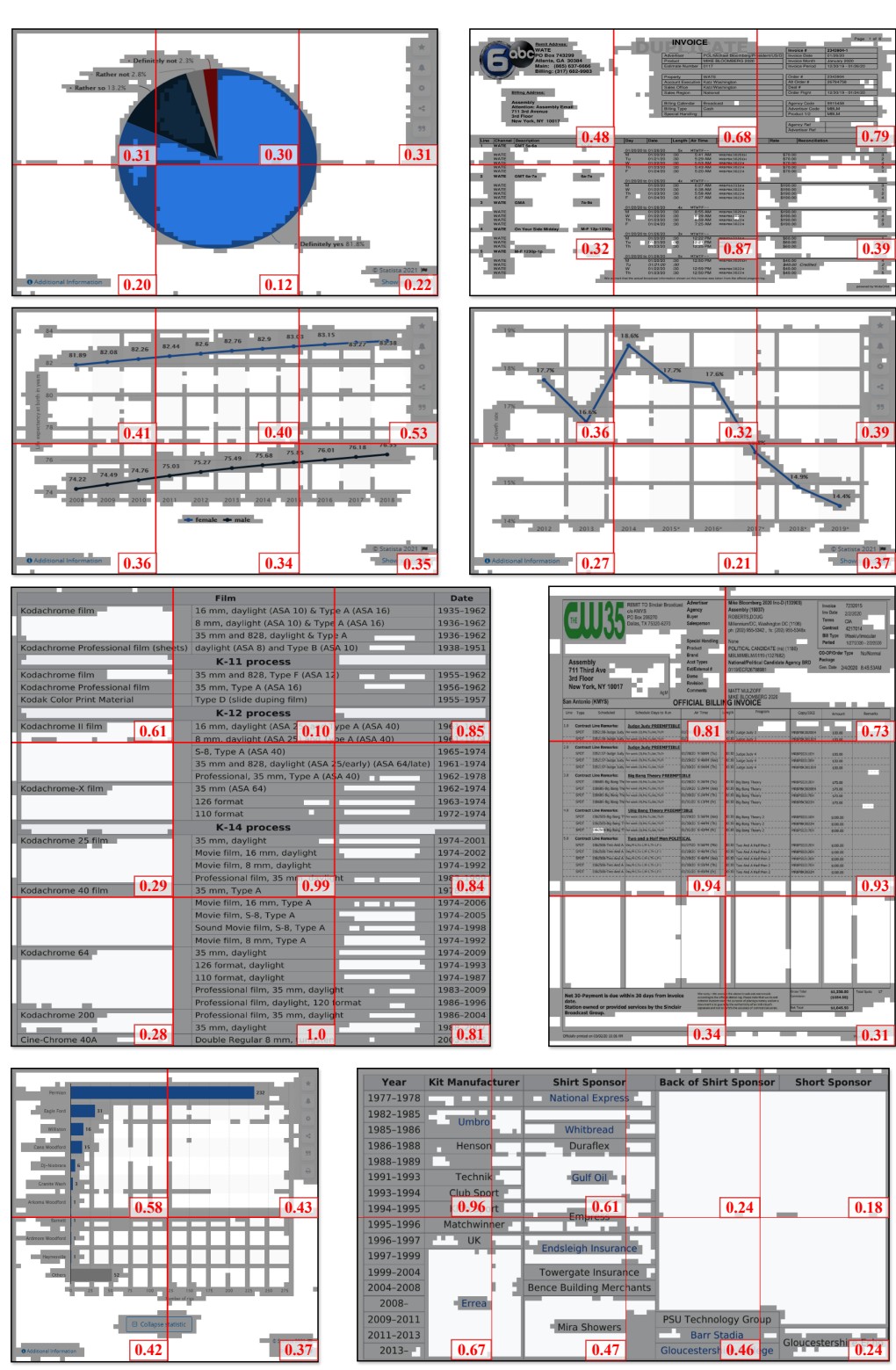

Figure 14: More visualization results of identified redundant tokens in additional datasets with various distributions.

### A.3.3 SELECTED INFORMATIVE TOKENS VISUALIZATION

In Figure 15, there are more results regarding the informative tokens selected with the guidance of `CLS`-patch correlation.

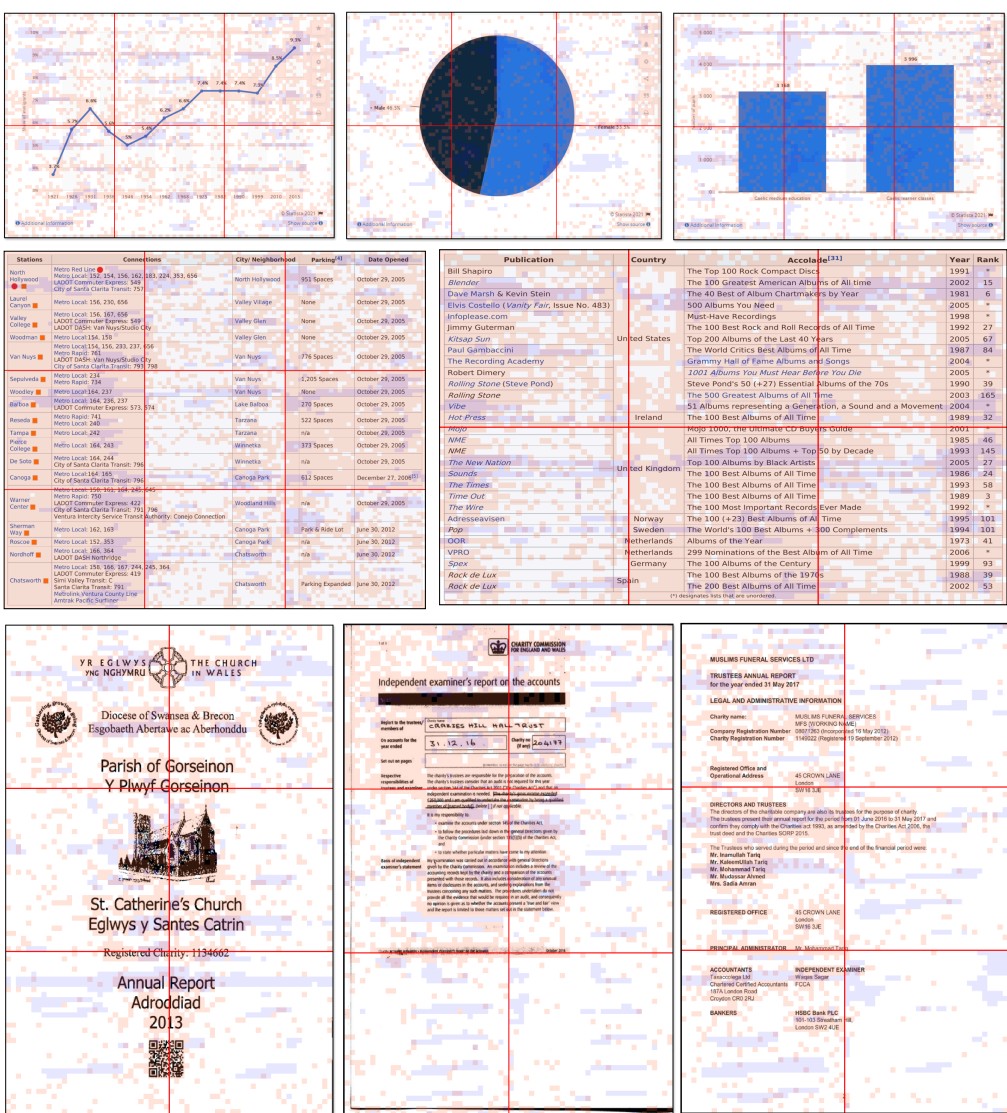

Figure 15: More visualizations of `CLS`-patch correlation-based token selection across various datasets with different distributions.

## B  BROADER IMPACT AND POTENTIAL RISK

Our approach employs readily available Multimodal Large Language Models (MLLMs), which means it shares some of their limitations, including the production of biased results. We recommend thoroughly evaluating its safety and fairness for the intended use before applying this method in practice.

