# OpenReview forum: "Token-level Correlation-guided Compression for Efficient Multimodal Document Understanding"
_ICLR.cc/2025/Conference — ICLR 2025 Conference Withdrawn Submission_

### Official Review · Reviewer_2zFK · 2024-10-27

**Soundness:** 2
**Presentation:** 3
**Contribution:** 2
**Rating:** 5
**Confidence:** 4

**Summary:**

This paper focuses on document understanding. Most current methods ignore the different informativeness of the tokens in the sub-images, leading to a significant increase in the number of image tokens. Hence, the authors propose a plug-and-play Token-level Correlation-guided Compression module, which evaluates the repeatability and informativeness based on the correlation between image tokens. Experiments demonstrate the effectiveness of this module.

**Strengths:**

Topics are current and important, focusing on compressing redundant information to improve model inference efficiency. Experimental results show that the proposed method outperforms existing solutions in terms of token compression efficiency and performance retention.

**Weaknesses:**

1) The role of the prompt is overlooked. Given that token compression is based entirely on images, it remains unclear how to ensure that the information required for the prompt is retained. It would be better to compare CLS patch-based and prompt-based approaches to token compression guidance.
2) The effectiveness of the plug-and-play approach requires further validation. It would be beneficial to demonstrate results on more baselines.
3) The analysis of experiments needs to be enriched. For example, why the model fine-tuned on TextVQA, TextCaps and VisualMRC datasets is not as good as plug-and-play instead.
4) There is an absence of direct comparisons with other token compression methods. It would be better to evaluate the performance and efficiency of the proposed method alongside existing token compression methods under consistent baselines.
5) The effectiveness of the Global Info Token requires additional validation. As shown in Figure 9, the Global Info Token is not consistently positioned in information-rich areas. Expanding the results presented in Table 4 to include more datasets could provide a more comprehensive assessment.

**Questions:**

Compared to PruMerge+, what are the advantages of the token compression method proposed in this paper?

---

### Official Review · Reviewer_LVBq · 2024-11-01

**Soundness:** 3
**Presentation:** 2
**Contribution:** 2
**Rating:** 5
**Confidence:** 5

**Summary:**

This paper focus on the token compression for Multimodal Large Language Models. The authors propose Token-level Correlation-guided Compression. By calculating the information density of sub-images and efficiently capturing the most informative tokens through the correlation between the [CLS] token and patch tokens, the authors have developed a plug-and-play compression module to remove redundant tokens produced by existing image cropping methods, accelerating the model's training and inference speed.

**Strengths:**

The Token-level Correlation-guided Compression filters out the truly informative parts from the visual tokens through patch-patch correlation-guided information density calculation and cls-patch correlation-guided informative token sampling, accelerating the model's training and inference speed.

**Weaknesses:**

1.Although the authors' method compresses visual tokens and improves the speed of model training and inference, it results in a noticeable performance drop, and the speed improvement is not outstanding.

2.The author claims that the proposed method is plug-and-play, but they only conducted experiments on the DocOwl-1.5 model, which is not very convincing. The proposed method needs to be validated on more models, especially on some higher-performance models such as InternVL2 and QWEN2VL.

3.Lacking comparison with some of the latest token compression methods, such as FastV, which has been accepted by ECCV.

Chen L, Zhao H, Liu T, et al. An image is worth 1/2 tokens after layer 2: Plug-and-play inference acceleration for large vision-language models[J]. arXiv preprint arXiv:2403.06764, 2024.

4.Lacking results on the commonly used benchmark OCRBench.

Liu Y, Li Z, Yang B, et al. On the hidden mystery of ocr in large multimodal models[J]. arXiv preprint arXiv:2305.07895, 2023.

5.In Line 139, the authors claim that "Despite their robust capabilities for document understanding, these models remain significantly inefficient." However, in the experimental tables, the author does not compare the efficiency with other methods. It would be better to include a comparison of efficiency with other methods in Table 1.

**Questions:**

Does compressing tokens also offer advantages in terms of memory consumption? If so, the author should present this information. The proposed method does not seem to be designed specifically for documents, so why were experiments only conducted on document understaning benchmarks? Why not conduct experiments on more general multimodal understanding as well?

---

### Official Review · Reviewer_oT3k · 2024-11-03

**Soundness:** 3
**Presentation:** 3
**Contribution:** 2
**Rating:** 3
**Confidence:** 5

**Summary:**

This paper proposes a Token-level Correlation-guided Compression (TCC) method for dynamic visual token compression, aiming to improve the efficiency of Multimodal Large Language Models (MLLMs) in document understanding tasks. The method compute the information density through patch-patch correlations, leveraging token correlations to guide the compression process in document understanding tasks, aiming to reduce the number of visual tokens while maintaining model performance.  Experiments with mPLUG-DocOWL1.5 shows the effectiveness of the proposed method.

**Strengths:**

The paper introduces the idea of information density based on patch-patch correlations, defining the information density of sub-images as the proportion of non-redundant tokens. This is  then used to determine the token compression ratios adaptively. The proposed idea  has some technical values.

**Weaknesses:**

1. The innovation of the proposed method is limited and not strong. Analyzing correlations or importance between visual tokens for compression of vision tokens has been explored in many recent researches. The simple idea of analyzing correlations between image patches and using [CLS] token correlations with patch tokens to sample the most informative tokens is straightforward and doesn't bring many new insights to the field.

2. In Table 1, the paper only compares with two token compression methods, which is not sufficiently convincing. There are several recent dynamic visual token compression methods, but the authors did not compare with these methods, such as FastV and more:
   (1)  Chen, L. et al.,  An image is worth 1/2 tokens after layer 2: Plug-and-play inference acceleration for large vision-language models. arXiv preprint arXiv:2403.06764.
   (2)  Lin, Z.; Lin, M.; Lin, L.; and Ji, R. 2024. Boosting  Multimodal Large Language Models with Visual Tokens Withdrawal for Rapid Inference arXiv preprint arXiv:2405.05803.
   (3). Zhang J. et al., Dockylin: A large multimodal model for visual document understanding with efficient visual slimming, arXiv preprint arXiv:2406.19101
    ...

3. While the authors claim their method is plug-and-play and can be seamlessly integrated into existing MLLMs using the proposed techniques, they only validated it on mPLUG-DocOWL1.5. The lack of experiments on other document understanding multimodal models (especially some recent SoTA models) makes it unclear whether the method is universally effective,  or just works well for mPLUG-DocOWL1.5. The experiments are not  solid and convincing enough.

4. The compression effect is limited, achieving only 11.5% compression on mPLUG-DocOWL1.5. This compression ratio is not impressive compared to other recent multimodal model token compression methods.

**Questions:**

--  In Section A.2.3, the paper describes the results of Local Information Mining when selecting different layers. How does the performance of Global Information Mining vary with the selection of different layers?

---

### Official Review · Reviewer_6yhF · 2024-11-05

**Soundness:** 2
**Presentation:** 2
**Contribution:** 1
**Rating:** 3
**Confidence:** 4

**Summary:**

The paper proposes multiple heuristics to improve the speed of Multi-Modal Large Language Models for document understanding. This is achieved by discarding uninformative tokens before feeding into the MLLM. For this purpose, the paper also comes up with a heuristic definition of "information density". The proposed method speeds up one of the state of the art models (DocOW11.6) by 13% in average (in exchange of minor performance degradation).

**Strengths:**

- Two simple and intuitive heuristics providing non-trivial speed improvements for MLLM models.
- Multiple ablations are performed to justify the design decisions

**Weaknesses:**

- The paper proposes "incremental" (13%) efficiency improvements for the state of the art MLLM document understanding models in exchange for non-zero model accuracy regressions.
- The method requires fine-tuning of target MLLM model for the best results. Otherwise, the trade-off between model accuracy loss and inference speed improvement is not justifiable.
- Theoretical justification for the proposed "information density" method is absent.
- Limitations of the proposed method needs to be discussed in depth.

**Questions:**

Q1: Figure 8 and 9 suggests that the proposed method is primarily good at identifying the background pixels as uninformative. Would document background detection perform better as a baseline to detect and remove uninformative patches before feeding into MLLM?
Q2: Can the proposed method be applied beyond document understanding? Is there an overarching theme?

---

### Note · Authors · 2024-11-13

**Comment:**

I have read and agree with the venue's withdrawal policy on behalf of myself and my co-authors.

**Withdrawal Confirmation:**

I have read and agree with the venue's withdrawal policy on behalf of myself and my co-authors.